# Resonant Magnetoelectric Coupling of Fe-Si-B/Pb(Zr,Ti)O_3_ Laminated Composites with Surface Crystalline Layers

**DOI:** 10.3390/s23249622

**Published:** 2023-12-05

**Authors:** Yu Sun, Xu Zhang, Sheng Wu, Nian Jiang, Xin Zhuang, Bin Yan, Feng Zhang, Christophe Dolabdjian, Guangyou Fang

**Affiliations:** 1Aerospace Information Research Institute, Chinese Academy of Sciences, Beijing 100094, China; sunyu21a@mails.ucas.ac.cn (Y.S.); zhangxu2019@imu.edu.cn (X.Z.); yanbin@aircas.ac.cn (B.Y.); zhangfeng002723@aircas.ac.cn (F.Z.); gyfang@mail.ie.ac.cn (G.F.); 2University of Chinese Academy of Sciences, Beijing 100049, China; 3School of Physical Science and Technology, Inner Mongolia University, Hohhot 010021, China; 4Yantai Research Institute of Harbin Engineering University, Harbin Engineering University, Harbin 264006, China; wusheng@hrbeu.edu.cn (S.W.); jiangnian365@gmail.com (N.J.); 5Normandie Univ., UNICAEN, ENSICAEN, CNRS, GREYC, Bd Maréchal Juin, 14000 Caen, France; christophe.dolabdjian@unicaen.fr

**Keywords:** magnetoelectric effect, surface modification, magnetostriction

## Abstract

The resonant magnetoelectric (ME) effect of Fe_78_Si_9_B_13_/Pb(Zr,Ti)O_3_ (FeSiB/PZT) composites with a surface-modified Fe_78_Si_9_B_13_ amorphous alloy has been studied. The surface-modified FeSiB can improve the ME coefficient at the resonant frequency by optimizing the magnetomechancial power conversion efficiency. The maximum ME coefficient of the surface-modified ribbons combined with soft PZT (PZT5) is two-thirds larger than that of the composites with fully amorphous ribbons. Meanwhile, the maximum value of the ME coefficient with surface-modified FeSiB ribbons and hard PZT (PZT8) is one-third higher compared with the fully amorphous composites. In addition, experimental results of magnetomechanical coupling properties of FeSiB/PZT composites with or without piezoelectric layers indicate that the power efficiency of the composites first decreases and then increases with the increase in the number of FeSiB layers. When the surface crystalline FeSiB ribbons are combined with a commercially available hard piezoelectric ceramic plate, the maximum magnetoelectric coupling coefficient of the ME composite reaches 5522 V/(Oe*cm), of which the electromechanical resonant frequency is 23.89 kHz.

## 1. Introduction

Fe-based amorphous alloy, also known as Fe-based metallic glass, has been widely used in laminated magnetoelectric (ME) composites and devices because of its excellent magnetomechanical properties [1,2,3,4]. This type of composite can realize the mutual conversion between magnetic and electric energy through the strain or stress between the magnetostrictive and piezoelectric materials. The devices made from ME composite material include magnetic field sensors, miniaturized antennas, and high-efficiency energy converters [5,6,7,8,9]. In order to meet the main technical indexes of these devices in practical application, the ME composite materials need to have strong ME coupling effects at room temperature. This usually requires improving the magnetostrictive properties of magnetic materials in ME composites, and then requires high-performance magnetostrictive materials and piezoelectric materials as single-phase materials in ME composites [10,11,12,13,14]. Therefore, optimization of the Fe-based amorphous materials in ME composites is an important step to carry out the subsequent research, which has also attracted the attention of researchers in recent years [15,16,17,18].

Electromagnetic waves with lower carrier frequencies have less propagation attenuation in lossy media, so very low frequency (VLF) antennas can solve the attenuation problem of high-frequency antenna transmission in high-loss environments [19,20,21]. However, the antenna needs to match the size of the wavelength to achieve high transmission efficiency, which leads to the large size and poor maneuverability of the traditional VLF transmitting antenna. The acoustic excitation antenna made of ME composite material is a kind of antenna that uses mechanical vibration to transmit low-frequency magnetic field signals [22,23,24,25]. The energy storage and conversion mechanism of this antenna is quite different from that of traditional antennas. Because the wavelength of sound waves is smaller than that of electromagnetic waves, energy can be stored in the form of kinetic energy or potential energy during the VLF mechanical motion of the antenna. Meanwhile, the mechanical strain/stress drives the magnetic dipoles inside the magnetostrictive material to transmit electromagnetic fields. Therefore, the resonant antenna based on ME composite material can achieve higher electromagnetic wave transmission efficiency in the VLF band with a smaller antenna volume.

Ivasheva et al. [18] used another Fe-based amorphous alloy (AMAG493) and a PZT-19 piezoelectric plate to make a magnetostrictive/piezoelectric heterostructure device with a size of 30 mm × 10 mm. The resonant frequency of the ME sample is 53.4 kHz, and the maximum magnetoelectric coupling coefficient is 29.52 V/(Oe*cm). This work also focuses on the ME effect of composites, which consist of a PZT ceramic plate and FeSiB amorphous alloy ribbon. The further expansion of the magnetoelectric coupling coefficient ($\alpha_{ME}^{V}$) formula under electromechanical resonant frequency (EMR) shows that its value is closely related to the magnetomechanical coupling factor (*k*) and quality factor (*Q*) of magnetostrictive material when the thickness ratio (*n*) of the magnetostrictive layers to the composite is determined. In an energy conversion system, the *k*^2^ value [26] represents the ratio of conversion power to storage power, while the *Q* value [27] represents the storage power to loss power ratio. Thus, the product of the two factors, *k*^2^*Q*, is defined as the ratio between the conversion power and the loss power. Heat treatment on Fe-based amorphous ribbons at the proper temperature can effectively improve this efficiency factor, *k*^2^*Q*, according to our previous studies [28,29]. We found that the $\alpha_{ME}^{V}$ value of the ME composite prepared from the surface-modified FeSiB amorphous alloy can be significantly improved under EMR. The maximum magnetoelectric coefficient with surface crystalline FeSiB and hard piezoceramic PZT8 can reach 5522 V/(Oe*cm).

## 2. Experiment

A series of FeSiB ribbons with the dimensions of 80 mm × 3 mm × 0.025 mm were annealed in an air atmosphere under 440 °C/500 °C for 20 min and cooled down to the ambient temperature in the air. The FeSiB laminates of different thicknesses were made with the help of annealed FeSiB ribbons and epoxy resin. Soft piezoelectric ceramics PZT5 (Yu Hai Electronic Ceramics Co., Ltd., Zibo, China) with the dimensions of 40 mm × 3 mm × 0.5 mm and hard PZT ceramics PZT8 with the dimensions of 40 mm × 3 mm × 0.8 mm were selected as piezoelectric phase materials. Each two PZT pieces were connected head to end to form one PTZ layer, and then two FeSiB layers with the same thickness were laminated under the action of epoxy resin and hot-pressing machine to constitute a magnetostrictive/piezoelectric/magnetostrictive trilayer heterogeneous structure with the dimensions of 80 mm × 3 mm. The schematic diagram of the heterogeneous structure is given in Figure 1.

Three heterostructures were hot-pressed for more than 24 h. After that, the magnetostrictive layers were fully bonded with the piezoelectric layer, the samples were inserted into a 30 cm long winding solenoid coil that has an inductance value of 7.5 mH at 1 kHz. The *k* and *Q* values were measured by an impedance analyzer (HP 4294A) using resonant and anti-resonant methods [30,31]. Subsequently, the prepared FeSiB/PZT/FeSiB composites, along with the winding coil, were placed in the center of a pair of permanent magnets. According to the previous studies, the magnetic bias field (*H_dc_*) generated by permanent magnets was adjusted to maximize the *k* value. Impedance analyzer was used to test the impedance curve vs. frequency and to determine the EMR frequency (*f_r_*).

The ME voltage coefficient ($\alpha_{ME}^{V}$) is a measure of the voltage that is induced by an applied ac magnetic field *H*. Therefore, larger $\alpha_{ME}^{V}$ value means greater electric field intensity that is caused by a unit magnetic field intensity. Voltage source (Wave Factory, WF1968), low noise voltage amplifier (Stanford Research Systems, Model SR560), oscilloscope (Tektronix MSO46), and Helmholtz coil (PS-1HM365) were used to test $\alpha_{ME}^{V}$ values in the experiment. The test circuit diagram is detailed in Figure 2.

The conversion factor of the Helmholtz coil was 10 nT/mA, which provides a magnetic field along the length direction of the samples. The frequency of the voltage source was adjusted close to the EMR of the composite, and the composite was put in the center of the Helmholtz coil together with the solenoid coil. Two permanent magnets were placed at both ends of the composite along the axial direction, and the distance between the permanent magnets and the composite was changed to adjust *H_dc_*, while the oscilloscope was observed. A gain of five was set for the low noise voltage amplifier. When the output voltage *V_out_* reaches the maximum value, the position of permanent magnets is fixed while the value of frequency sweeping near *f_r_*. The input voltage *V_in_* measured across the resistor *R_m_* and the output voltage *V_out_* measured from the piezoelectric ceramic of the composite were recorded. The values of $\alpha_{ME}^{V}=\frac{V_{out}}{H}$ (or $\alpha_{ME}^{E}=\frac{V_{out}}{t_{p}H}$, where $t_{p}$ is the thickness of piezoelectric layer.) were calculated from the ratio between the input field and the output voltage as a function of the sweeping frequency.

## 3. Results

According to our previous investigations in [29], the *k* value of the FeSiB ribbon reaches the maximum when the sample is annealed at 440 °C for 20 min. However, a much higher annealing temperature is required for the efficiency factor *k*^2^*Q* to reach the maximum value. Two commercially available piezoelectric materials, PZT5 and PZT8, were combined with FeSiB ribbons that have been annealed in air at 440 °C and 500 °C for 20 min. As such, four ME composites with different combinations were prepared: 440 °C-FeSiB/PZT5, 440 °C-FeSiB/PZT8, 500 °C-FeSiB/PZT5, 500 °C-FeSiB/PZT8. The thickness values of PZT5 and PZT8 were 0.5 mm and 0.8 mm, respectively. On both sides of PZT5, the associated layer number of the FeSiB is six, while on both sides of the PZT8, there are nine layers of FeSiB ribbons. The ratio between the thickness of the FeSiB laminates (including top and bottom) to the total thickness of the composite is 0.38.

The values of $\alpha_{ME}^{E}$ for two kinds of FeSiB/PZT5 composites were plotted as a function of the frequency, as shown in Figure 3. For the FeSiB ribbons annealed at 440 °C for 20 min, the maximum magnetoelectric coupling coefficient $\alpha_{MEmax}^{E}$ of 440 °C-FeSiB/PZT5 reaches 598 V/(Oe*cm) at the resonance and the corresponding *f_r_* is 22.2 kHz. When the FeSiB ribbons were replaced by those annealed at 500 °C for 20 min, the value of $\alpha_{MEmax}^{E}$ of 500 °C-FeSiB/PZT5 was increased to 1025 V/(Oe*cm) at 22.34 kHz, which is 71% higher than the one with FeSiB annealed at 440 °C.

In Figure 4a, for the continuously stacking magnetostrictive layer, the $\alpha_{ME}^{E}$ of FeSiB/PZT5 composite increases at first and then decreases, which is also in line with the prediction of the ME equivalent circuit model [32,33]. It can also be observed in Figure 4 that the resonant frequency scales up with the increase in the magnetostrictive layer’s thickness. It can also be observed in Figure 4 that when soft PZT5 is selected as the piezoelectric layer, six layers of FeSiB on top and six layers on bottom are required to maximize the magnetoelectric coefficients.

Using the same experimental method, the values of $\alpha_{ME}^{E}$ were measured as a function of the frequency for FeSiB/PZT8 heterostructures with different thickness values of magnetostrictive layers, as shown in Figure 5.

It can be seen that when FeSiB is annealed at 440 °C for 20 min, the maximum value of $\alpha_{ME}^{E}$ of heterostructure with nine FeSiB layers is 4311 V/(Oe*cm) at 23.76 kHz. When FeSiB is annealed at 500 °C for 20 min, the $\alpha_{MEmax}^{E}$ of the ME composite with nine FeSiB layers and PZT8 is increased to 5522 V/(Oe*cm), which is 28% higher than the sample 440 °C-FeSiB/PZT8. The EMR frequency increases slightly to 23.89 kHz. According to the data in Figure 3 and Figure 5, we could obtain the same experiment phenomenon, that is, $\alpha_{ME}^{E}$ of the composite annealed at a higher temperature is larger than that was annealed at a lower temperature under the condition that the thickness of the magnetostrictive layer is unchanged. Meanwhile, the α-Fe(Si) crystallites have been confirmed to appear on both sides of the ribbons, when the ribbons are annealed at 500 °C for 20 min, following the X-ray diffraction measurements in ref [29]. The surface crystallization layer formed by high-temperature annealing is shown to effectively improve the magnetoelectric coupling ability in the ME composite. Moreover, when FeSiB ribbons are annealed at 500 °C, its magnetomechanical efficiency factor (*k*^2^*Q*) is also optimized following our previous investigations [29], as discussed in later sections.

The resonant frequency and the maximum $\alpha_{ME}^{E}$ of FeSiB/PZT8 composite with the increase in FeSiB layer number are also shown in Figure 6. The trend of variation with the resonant frequency in Figure 6 is similar to that in Figure 4. The resonant frequency increases with the increase in the thickness of FeSiB layer. However, the $\alpha_{ME}^{E}$ coefficient increases at first and then decreases with the increase in the FeSiB thickness value. When the hard PZT8 is selected as the piezoelectric layer, it requires nine layers FeSiB on both the top and bottom to maximize the $\alpha_{ME}^{E}$ coefficient.

By comparing the values of $\alpha_{ME}^{E}$ of two kinds of ME devices, the magnetoelectric coupling ability of the composite after surface modification treatment is effectively improved. This also provides the possibility for us to further use this device to prepare efficient transducers, magnetoelectric antennas and so on. Using this device can ensure that high-efficiency energy transmission can be achieved while maintaining a small size, so as to achieve the goal of a miniaturized low-frequency antenna.

According to the above discussion, the magnetoelectric coupling coefficient $\alpha_{ME}^{E}$ of FeSiB/PZT5 composite annealed at 440 °C is the lowest, and that of FeSiB/PZT8 composite annealed at 500 °C is the highest in the four groups of experimental data. We compared magnetomechanical properties of these two samples before and after adding piezoelectric layer. Experiment results are shown in Figure 7 and Figure 8.

Figure 7 shows the variation of the magnetomechanical coupling coefficients *k* of the FeSiB/PZT5 composites with the increase in the FeSiB layer number. All the samples in Figure 7 are made with the same PZT5 ceramic plate by changing the different FeSiB layers attached to ceramic plates. The experimental results in Figure 7a show that the thin FeSiB layer will result in a small *k* value for the epoxy-FeSiB composite, but with the increase in the thickness of the magnetostrictive layers, the *k* value of the ME composite gradually increases until the *k* value of the epoxy-bonded FeSiB composite is exceeded by that of the FeSiB/PZT5 composite. There are two main reasons for this. On the one hand, when the thickness of magnetostrictive layer increases, the demagnetization factor reduces the *k* value [34]. On the other hand, with the addition of piezoelectric layer, the average compliance coefficient (*s*) of the composite decreases, thus the *k* value of the composite increases when the piezomagnetic coefficient (*d_m_*) and magnetic permeability (*μ*) remain basically unchanged according to the calculation formula $k^{2}=\frac{d_{m}^{2}}{\mu s^{H}}$. *Q* value of the ME composites increases significantly compared with that of the single magnetostrictive layer, as shown in Figure 7b. Under the combined action of the magnetomechanical coupling coefficient *k* and quality factor *Q*, the efficiency factor *k*^2^*Q* of the ME composite decreases first and then increases compared with the epoxy-bonded FeSiB composite, as shown in Figure 7c. It can be seen that the introduction of the piezoelectric layer can indeed improve the energy transmission efficiency of this heterogeneous structure.

Similarly, the variation of the measured magnetomechancial coefficients of FeSiB/PZT8 composites with the layer numbers is shown in Figure 8. The variation trend of the *k*, *Q*, *k*^2^*Q* curves resemble to those in Figure 7, the efficiency factor *k*^2^*Q* of the composites first decreases and then increases with the increase in the number of FeSiB layers.

## 4. Discussion

Previous studies [28,29] have shown that annealing at an appropriate temperature can improve the magnetomechanical properties of the Fe-based amorphous ribbon. Therefore, we used the annealed FeSiB ribbons to fabricate the ME-laminated composites and tested their magnetomechanical properties in this experiment. Following the ME equivalent circuit model in Refs. [33,35], the ME coefficient $\alpha_{ME}^{V}$ of the ME composites at resonant frequency can be expressed in terms of piezoelectric ($d_{p}$), piezomagnetic ($d_{m}$), compliance coefficient ($s^{E}$, $s^{H}$) and quality factor (Q), as shown in Equation (1),
(1)$$\alpha_{ME}^{V}={\left|\frac{dV}{dH}\right|}_{r}=\frac{8}{\pi^{2}}Q\frac{nt_{p}d_{m}g_{p}}{ns^{E}+\left(1-n\right)s^{H}}.$$

The formula is further simplified by considering the composite compliance coefficient of the composite material, given as
(2)$$1/\bar{s}=n/s^{H}+\left(1-n\right)/s^{E},$$
where $s^{E}$ is the compliance coefficient of piezoelectric layer, and $s^{H}$ is the compliance coefficient of magnetostrictive layer. Substituting into Formula (1), we have
(3)$$\left\{\begin{array}{c} \alpha_{ME}^{V}={\left|\frac{dV}{dH}\right|}_{r}=\beta \frac{8}{\pi^{2}}Qnt_{p}d_{m}g_{p}\\ \beta =\frac{\bar{s}}{s^{H}s^{E}} \end{array}\right.,$$
where the piezomagnetic coefficient is deduced as $d_{m}=k\sqrt{\bar{s}\mu }$. Similarly, the piezoelectric charge constant is expressed as $d_{p}=k_{p}\sqrt{s\varepsilon }$, and the piezoelectric voltage constant is $g_{p}=\frac{d_{p}}{\varepsilon }$. *β* represents the composite compliance coefficient parameter. Substituting the above two equalities into Equation (3) yields
(4)$$\alpha_{ME}^{V}={\left|\frac{dV}{dH}\right|}_{r}=\beta \bar{s}\frac{8}{\pi^{2}}\sqrt{\frac{\mu }{\varepsilon }}n\left(1-n\right)tQkk_{p}.$$

Equation (4) summarizes the relation between the magnetoelectric coupling coefficient of the composite and *k* and *Q* values. The magnetomechanical properties of four kinds of ME composites calculated from the verified Equation (4) are measured, as given in Table 1.

Figure 9 shows the comparison of the calculated and measured values of the magnetoelectric coupling ability of four kinds of composites in Table 1. Due to the limitation of measuring means, the variation of some parameters in Equation (4), such as permeability and the compliance coefficient with magnetostriction, cannot be accurately measured, so we adopt a semiquantitative method to convert the parameters that cannot be accurately measured into measurable parameters by using the proportional relationship. In Equation (4), the permeability μ and dielectric constant ε are directly proportional to the inductance *L* and relative dielectric constant $\varepsilon_{r}$, respectively, according to $L=\frac{k_{L}\mu_{0}\mu N^{2}S}{l}$, where *N* represents the number of turns of the solenoid coil, *S* indicates the cross-sectional area of the coil, *k_L_* depends on the ratio of *S* to the length of the coil *l*, and $\varepsilon =\varepsilon_{r}\varepsilon_{0}$. $\frac{8}{\pi^{2}}$ and thickness ratio *n* are constants, so the calculation formula of the relative value of the ME coefficient can be expressed as Cal.=LNεrtQkkp. Accordingly, the unit of relative values derived from this formula is $\sqrt{H}\frac{m}{s}$, which does not represent any actual physical meaning. For this reason, it is impossible to compare the calculated relative values directly with the measured values. However, the results in Figure 9 show that the variation trend of the calculated relative values of the composites is highly consistent with the measured values. When the piezoelectric material is the same, the composite with higher temperature annealed FeSiB has a higher ME coefficient at EMR, while the FeSiB material is the same, the ME coefficient of the composite with hard PZT8 is also higher.

The Fe-based amorphous ribbon prepared by the single-wheel rapid quenching process has an uneven distribution of atomic chemical components along the thickness direction. The composition of metalloid atoms on the surface of the ribbon is often less than that of the inner region, so the crystallization temperature near the ribbon surface is lower. After the annealing process at an appropriate temperature for a certain period of time, the surface of the Fe-based amorphous ribbon is partially crystallized, while the internal state remains amorphous. This is a heterostructure composed of surface crystalline and amorphous reminders, which can significantly improve the ME coefficient of Fe-based amorphous ribbons when the ribbons are excited at resonant frequency. The surface modification of FeSiB ribbons after annealing reduces the magnetic loss by establishing the surface-to-interior stress. The ME coefficients for the composites with the annealed FeSiB ribbons that are crystallized near the surface regions are enhanced compared to the amorphous ones.

## 5. Conclusions

FeSiB ribbons annealed at 440 °C and 500 °C, respectively, are bonded with two kinds of piezoelectric materials by epoxy to prepare ME composites. Measured values of $\alpha_{ME}^{E}$ show that Fe-based amorphous ribbons with surface crystallization can effectively improve the ME coefficients in magnetostrictive-piezoelectric heterostructures. The $\alpha_{ME}^{E}$ value of 500 °C-FeSiB/PZT8 heterostructure reaches 5522 V/(Oe*cm) at an EMR frequency of 23.89 kHz, which is approximately one-third higher than that for 440 °C-FeSiB/PZT8 composite. Furthermore, we compared the calculated and measured values of the ME coefficients by considering the measured inputs parameters, the results showed a good fit between calculation and measurement. Based on our results, we are able to conclude that the ME coefficient with surface-modified Fe-based amorphous alloy by the annealing process can be effectively improved at EMR frequency.

## Figures and Tables

**Figure 1 sensors-23-09622-f001:**
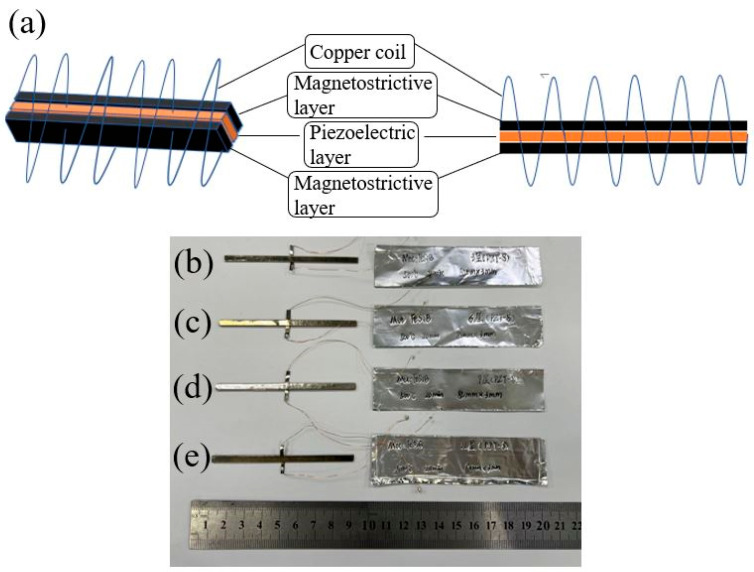
(**a**) Schematic diagram of magnetostrictive/piezoelectric/magnetostrictive heterostructures; physical patterns of (**b**) three-layer; (**c**) six-layer; (**d**) nine-layer and (**e**) twelve-layer magnetostrictive heterogeneous structures.

**Figure 2 sensors-23-09622-f002:**
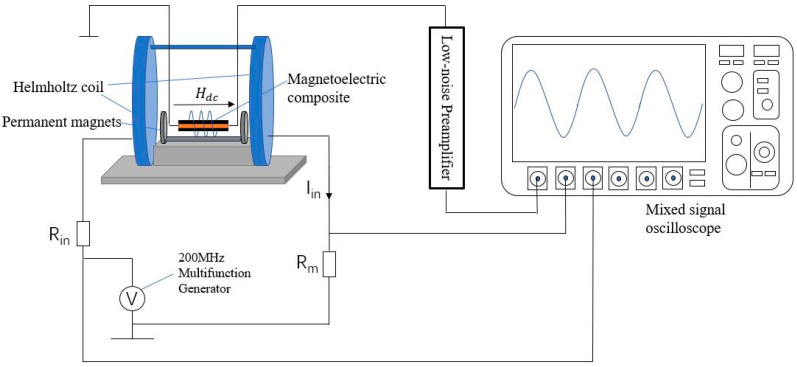
Diagram of the test circuit of magnetoelectric coupling coefficient.

**Figure 3 sensors-23-09622-f003:**
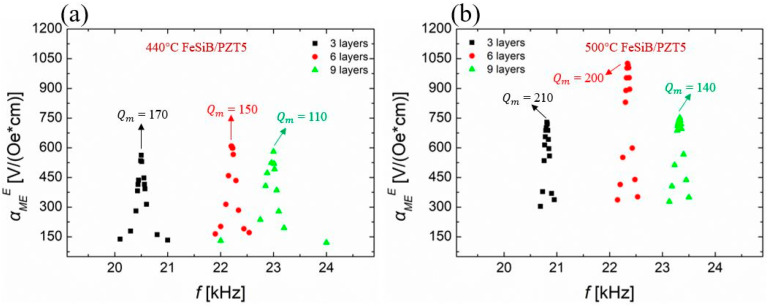
The values of $\alpha_{ME}^{E}$ for FeSiB/PZT5 composites with different layer number of FeSiB ribbons were plotted as a function of frequency. (**a**) FeSiB ribbons were annealed at 440 °C and (**b**) FeSiB ribbons were annealed at 500 °C.

**Figure 4 sensors-23-09622-f004:**
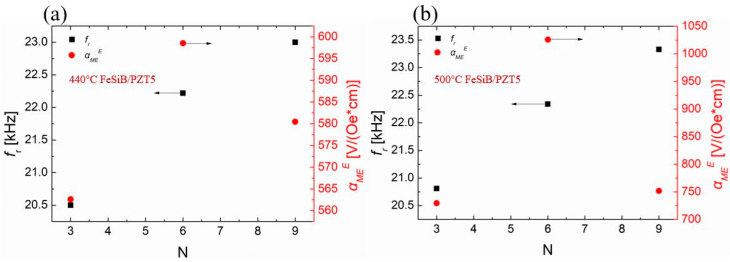
The EMR frequency and $\alpha_{ME}^{E}$ values of FeSiB/PZT5 composites as a function of the FeSiB layer number. (**a**) Composites of PZT5 and annealed FeSiB at 440 °C; (**b**) composites of PZT5 and annealed FeSiB at 500 °C.

**Figure 5 sensors-23-09622-f005:**
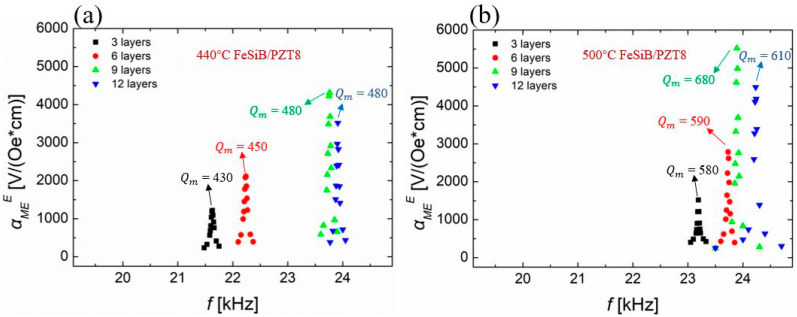
The values of $\alpha_{ME}^{E}$ of FeSiB/PZT8 composites with different thickness values of FeSiB as a function of frequency. (**a**) Composites of PZT8 and FeSiB annealed under 440 °C; (**b**) composites of PZT8 and FeSiB annealed under 500 °C.

**Figure 6 sensors-23-09622-f006:**
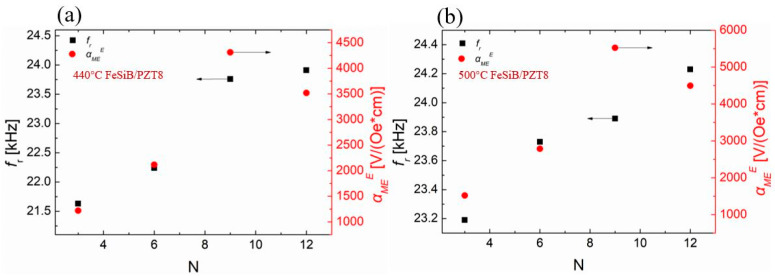
The EMR frequency and $\alpha_{ME}^{E}$ values of FeSiB/PZT8 composites as a function of the number of magnetostrictive layers. (**a**) Composites of PZT8 and FeSiB annealed under 440 °C; (**b**) composites of PZT8 and FeSiB annealed under 500 °C.

**Figure 7 sensors-23-09622-f007:**
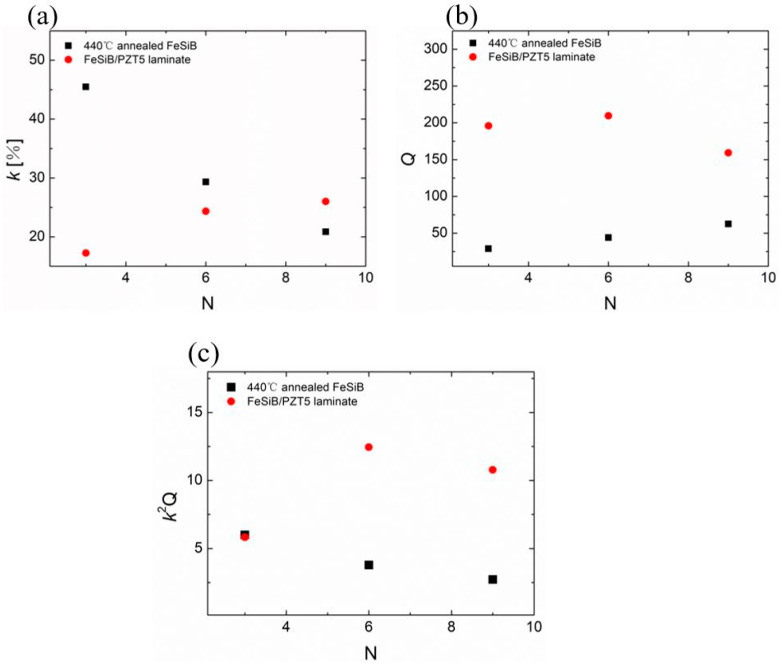
Magnetomechanical properties of FeSiB/PZT5 composites and of the corresponding FeSiB as a function of layer numbers. (**a**) The maximum value of *k* as a function of FeSiB layers. (**b**) The value of *Q* and (**c**) the value of *k*^2^*Q*, which correspond to the maximum *k* as a function of FeSiB layers.

**Figure 8 sensors-23-09622-f008:**
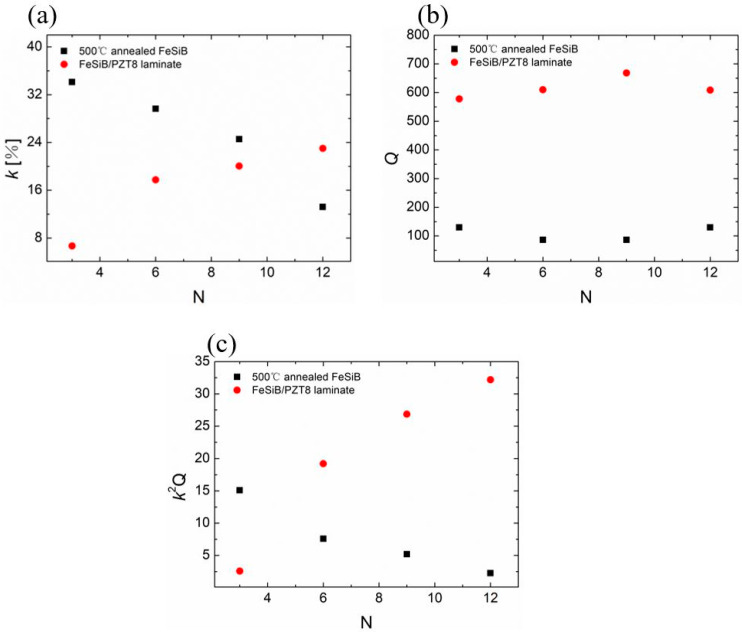
Magnetostrictive coefficients of FeSiB/PZT8 composite and corresponding magnetostrictive materials as a function of layer numbers. (**a**) The maximum value of *k* as a function of FeSiB layers. (**b**) The value of *Q* and (**c**) the value of *k*^2^*Q*, which correspond to the maximum *k* as a function of FeSiB layers.

**Figure 9 sensors-23-09622-f009:**
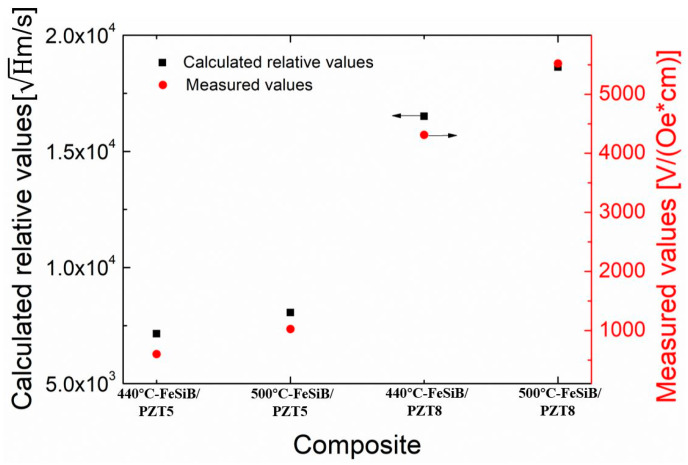
Comparison of the calculated and measured values of the magnetoelectric coupling capacity of four kinds of composites.

**Table 1 sensors-23-09622-t001:** Impact factors of $\alpha_{ME}^{V}$ value of composites.

Composites	N ^1^	*H_dc_* (Oe) ^2^	*L* (mH) ^3^	*Q*	*k* (%) ^4^	*k_p_* (%) ^5^	n ^6^	*T* (mm)	*f_r_* (kHz)	$\mathrm{Cal}.\;(\text{×}10^{3}\;\sqrt{H}\frac{m}{s}$) ^7^
440 °C-FeSiB/PZT5	6	9	49.3	150	25	29	0.38	0.8	22.20	7.2
500 °C-FeSiB/PZT5	6	9	44.1	200	24	27	0.38	0.8	22.34	8.1
440 °C-FeSiB/PZT8	9	13	52.0	480	26	24	0.38	1.3	23.76	16.5
500 °C-FeSiB/PZT8	9	13	50.1	680	22	23	0.38	1.3	23.89	18.6

^1^ N indicates the number of magnetostrictive layers; ^2^ the optimal values of the biased field corresponding to *k*. ^3^ the measure values of inductance(*L*) at 5 kHz; ^4^
*k* represents magnetostrictive end magnetomechanical coupling coefficient of the composite; ^5^
*k_p_* represents pizeoelectric end magnetomechanical coupling coefficient of the composite; ^6^ n represents the thickness ratio of magnetostrictive layer to the composite; ^7^ Cal. represents calculated relative values of $\alpha_{ME}^{V}$ as a function of the measured values of the impact factors in Equation (4).

## Data Availability

Data available on request due to restrictions.

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
