# Peer review of "Resonant Magnetoelectric Coupling of Fe-Si-B/Pb(Zr,Ti)O3 Laminated Composites with Surface Crystalline Layers"

_sensors, 2023, doi:10.3390/s23249622_

Round 1
Reviewer 1 Report
Comments and Suggestions for Authors
The article is devoted to the study of magnetoelectric properties of two-layer Metglass/PZT composites. Despite the interesting results, the article is poorly prepared and needs serious improvement:
1) The authors write about a giant ME effect in 5522 V/Oe*cm-1. I believe the authors made a mistake with the units of measurement, similar errors in the units of measurement in the figures.
2) The introductory part is presented very poorly, the motivation of this work is not given, and the selected material is better than those presented in the work. The clear novelty of the work is not presented
3) The list of references is very poorly presented. There are 21 links in total.
4) The authors write about surface crystalline Fe-Si-B, however, structural data are not provided that confirm this and there is no information about the microstructure of the composite.
Reviewer 2 Report
Comments and Suggestions for Authors
Manuscript number: sensors-2660359
Title: Resonant magnetoelectric coupling of Metglas/PZT laminated composites with surface crystalline Fe-Si-B
The present paper is interesting, however, I think that it needs some rectifications and additions before one can take a final decision. The specific comments are as follows:
1) The paper contains some grammatical errors and typo-mistakes that should be corrected.
2) The Abstract part could be further improved. It should clearly summarize the problem, state the concept and the method, and inform the important results and conclusions in the present study. Also, it should contain some qualitative and quantitative results.
3) The introduction part should be greatly improved. I feel that the authors can mention in the Introduction part about the fundamental and practical importance of the magnetoelectric composites. These materials are very promising materials for several practical applications, which can be highlighted in the Introduction part. However, in the introduction part, the authors did not sufficiently treat the importance/applications of these samples. So, some recent references should be inserted and discussed which will be very helpful for the researchers/readers.
4) The novelty, motivation, and objective in the present study are NOT well clear. The novelty statement of the given approach should be emphasized in the introduction. Also, the authors should re-elaborate the limitations of previous studies, the motivation behind the present work along with novelty statement, and the objectives of the current study at the end of the Introduction part.
5) Previous studies provided in the literature on Metglas/PZT composites are NOT sufficiently discussed in the Introduction part. The authors should report similar works that contain the properties of these composites, their limitations, and their hypothesis to improve the properties.
6) The following chapter “Advanced Progress in Magnetoelectric Multiferroic Composites: Fundamentals, Applications, and Toxicity” (https://doi.org/10.1007/978-3-030-90948-2_52) is very interesting, could be used, and several examples on diverse magnetoelectric composites could be extracted from it.
7) Why the magnetoelectric voltage is higher at higher temperatures?
8) The authors should provide some additional data/measurements such as piezoelectricity, ferroelectricity/polarization, dielectric, capacitance, etc.
9) X-ray photoelectron spectroscopy (XPS) analysis could be performed and discussed, which will be helpful for readers.
10) Morphology, XRD, Magnetization, and Raman could also be investigated.
Comments on the Quality of English LanguageThe paper contains some grammatical errors and typo-mistakes that should be corrected.
Reviewer 3 Report
Comments and Suggestions for Authors
Comments and Suggestions for Authors in the attached file.

Round 2
Reviewer 1 Report
Comments and Suggestions for Authors
Unfortunatelly the authors did not take into account my note:
2. Point-by-point response to Comments and Suggestions for Authors Comments 1: The authors write about a giant ME effect in 5522 V/Oe*cm-1. I believe the authors made a mistake with the units of measurement, similar errors in the units of measurement in the figures. Response 1: Thanks for your suggestion. We have modified the units to V/(Oe*cm) in the manuscript.
Here I mean that 5522 V/Oe*cm-1 looks unrealistic. Did you compare them with literature data? In discussion part I did not see any comparison with literature data...
Reviewer 2 Report
Comments and Suggestions for Authors
The revised manuscript has been improved. I think that it can be now accepted for publication.
Reviewer 3 Report
Comments and Suggestions for Authors
Comments and suggestions for authors in the attached file.

Round 3
Reviewer 1 Report
Comments and Suggestions for Authors
paper can be accepted
Reviewer 3 Report
Comments and Suggestions for Authors
Comments and suggestions for authors in the attached file.
